Neuromedin U and neurotensin may promote the development of the tumour microenvironment in neuroblastoma

Yang Daheng 1
Zhang Xianwei 1
Li Zheqian 1
Xu Fei 1
Tang Chenjie 2
Chen Hongbing 45088025@qq.com 1
1 Department of Clinical Laboratory, Children’s Hospital of Nanjing Medical University , Nanjing , China
2 Wuxi Children’s Hospital, Wuxi People’s Hospital Affiliated to Nanjing Medical University , Wuxi , China
Juan Hsueh-Fen
Electronic publication date: 2021 Jun 1
Publication date: 2021
Volume: 9
Electronic Location ID: e11512
Received 2020 Sep 14; Accepted 2021 May 4
Copyright: ©2021 Yang et al.
Copyright year: 2021
Copyright holder: Yang et al.
License: This is an open access article distributed under the terms of the Creative Commons Attribution License, which permits unrestricted use, distribution, reproduction and adaptation in any medium and for any purpose provided that it is properly attributed. For attribution, the original author(s), title, publication source (PeerJ) and either DOI or URL of the article must be cited.
License URL: https://creativecommons.org/licenses/by/4.0/

Keywords: Neuroblastoma, Tumour microenvironment, Neuromedin U, Neurotensin, INSS stage

Funding: The authors received no funding for this work.

==============================
Stage 4S neuroblastoma, as defined by the International Neuroblastoma Staging System committee (INSS), is known to regress spontaneously and have a more favourable outcome compared with stage 4 tumours. Comparing the molecular differences between these two stages may provide insights into the progression of neuroblastoma. Our study aimed to explore the molecular differences in the tumour microenvironment (TME) between INSS stage 4S and stage 4 tumours to provide an insight into the mechanisms underlying the biological processes of neuroblastoma. We downloaded the datasets GSE120572 and GSE73517 from the GEO database and pre-processed them using the limma package. CIBERSORT deconvolution agorithm was applied to analyse the differences in 22 infiltrating immune leukocyte subsets between the two stages. We used gene ontology (GO) enrichment analysis to determine the biological process (BP) annotation of differentially expressed genes (DEGs) using the online WebGestalt tool. Hub genes were determined in the STRING database and Cytoscape, and the expression of these genes was verified in the Oncomine database. Then these critical genes were performed survival analysis in TARGET database. We further validated the hub genes using a transwell assay and wound healing assay to detect the function of the genes in the neuroblastoma cell line SK-N-BE(2). GO analysis revealed that the 216 DEGs between stage 4S and stage 4 were enriched in aggressive biological processes. Neuromedin U (NMU) and neurotensin (NTS), which were significantly associated with patients’ overall survival rate, were verified to be elevated in stage 4, and to promote the proliferation and invasion of the SK-N-BE(2) cell. Tumour infiltrating leukocyte analysis showed a high infiltration of regulatory T cells and type 2 tumour-associated macrophages in stage 4 but not in stage 4S. Results of gene co-expression correlation, and the results of previous studies, suggest that NMU and NTS may play certain roles in modulating TME, thus facilitating the progression of neuroblastoma.

Introduction

Neuroblastoma arises during foetal or early post-natal life from the sympathoadrenal lineage of the neural crest and presents a broad clinical spectrum characterized by enormous biological and genetic heterogeneity. Neuroblastoma can spontaneously regress, mature, or demonstrate an aggressive and malignant phenotype that is poorly responsive to current multimodal therapy (Castleberry, 1997; Luksch et al., 2016). In what the International Neuroblastoma Staging System Committee (INSS) defines as stage 4, tumour cells metastasize to distant tissues via blood and lymphatic vessels, with obviously increased lymphangiogenesis. INSS stage 4S is a unique category of metastatic disease in infants with dissemination localizedin the liver and skin, with minimal marrow involvement (i.e., <10% of total nucleated cells identified as malignant by bone biopsy or by bone marrow aspirate) (PDQ Pediatric Treatment Editorial Board, 2002). Considering that stage 4S can regress spontaneously and has a favourable outcome compared with stage 4, we may be able to obtain insight into the progression of neuroblastoma by comparing the molecular differences between these two stages.

The tumour microenvironment (TME) is the cellular and matrix environment that includes tumour cells, immune cells, stroma cells, and the extracellular matrix (McCarthy, 2014). The imbalance between the effector and regulatory cell interactions in the TME are known to contribute to the major immune evasion mechanisms of tumours.Tumour-infiltrating lymphocytes (TILs) play a pivotal role in regulating tumour growth and anti-tumour progression. Researchers have confirmed that the infiltration of CD8+ T lymphocytes, CD4+ T lymphocytes, memory T lymphocytes, and natural killer (NK) cells is linked to a favourable outcome in several cancers including breast cancer, colorectal cancer, and neuroblastoma (Galon, Fridman & Pages, 2007; Mina et al., 2015). Conversely, a high level of immunosuppressive immune cells including regulatory T lymphocytes (Tregs), myeloid-derived suppressor cells (MDSCs), and tumour-associated macrophages type 2 (TAM-M2) may be involved in weakening the efficacy of anti-neoplasm immune responses (Carlson & Kogner, 2013) leading to aggressive tumour growth and an unfavourable outcome.

The molecular and cellular mechanisms underlying the interaction between immune cells and neuroblastoma have been studied. MYCN amplification has been determined to be independent high risk factor for stratification in neuroblastoma study (Luksch et al., 2016), however, the existence of key genes and the modulation of tumour microenvironment by these remain to be elucidated. In this study, we attempted to research the overall gene alteration between stage 4S and stage 4, and to explore the impact of key genes on TME processes to explain the mechanisms of neuroblastoma development.

Materials and Methods

Data pre-processing

Raw data matrices of GSE120572 (Ackermann et al., 2018) and GSE73517 (Henrich et al., 2016) datasets were obtained from the GEO database. These datasets were a natural source of neuroblastoma samples without additional confounding factors. The gene expression profiling conducted by array were based on the GPL16876 platform (Agilent-020382 Human Custom Microarray 44k (Feature Number version)). GSE120572 contained 46 INSS stage 4S and 168 stage 4 patients, and GSE73517 contained 20 stage 4S and 56 stage 4 patients. In GSE120572 dataset, stage 4 included 58 MYCN amplified and 109 non-amplified samples, and stage 4S included 6 amplified and 40 non-amplified samples. In GSE73517 dataset, stage 4 included 26 MYCN amplified and 30 non-amplified samples, and stage 4S included 3 amplified and 17 non-amplified samples. The clinical background of the datasets is shown in Table S1. We obtained the official gene symbol according to the probe ID files and duplicates were merged by averaging their expression. Neuroblastoma transcriptome expression data with clinical characteristics in TARGET (https://ocg.cancer.gov/programs/target) database were obtained for survival analysis of critical genes. In this dataset, stage 4 contained 27 MYCN amplified and 92 MYCN non-amplified samples. Stage 4S consisted of 1 MYCN amplified and 19 MYCN non-amplified samples (Table S2). All series expression matrices were extracted and normalized using R 3.6.1 software, limma package and survival package. Codes were uploaded to the Figshare platform (https://figshare.com/s/da446df5fa194bc24f90).

Statistical methods

The differentially expressed genes (DEGs) located between stage 4S and stage 4 were analysed based on Bayes’ test using the limma package and Pvalues were adjusted using the FDR method. The screening criteria of significant DEGs were set as —logFC— > 1 (FC means fold change of gene expression) and adj.P.Val < 0.05 (adjusted P value). Data from in vitro experiments were analysed using GraphPad Prism 8 software (San Diego, USA) with the unpaired t-test method. A significant difference was determined as P < 0.05.

Gene ontology enrichment analysis

To explore the overall function of specific genes, DEGs from the two datasets were intersected to obtain optimal differential genes. Gene Ontology (GO) enrichment analysis was used to explore the biological process (BP) annotation using the WebGestalt online tool (http://www.webgestalt.org). The basic parameters were set to include: Homo sapiens as the organism of interest, over-representation analysis (ORA) as the method of interest, gerontology as the functional database, and gene ID type as the gene symbol. Significant differences were set as FDR < 0.05.

Hub genes identification

The intersected genes were analysed in the STRING database (https://string-db.org/) to investigate the multiple protein interactions, with a 0.7 interaction score. Cytoscape software was used to visualize the networks. The intersection of the top 30 genes was ranked by degree and the hub genes were further identified using the plugin CytoHubba.

The expression values of the hub genes between stage 4S and stage 4 were obtained from the Oncomine database (https://www.oncomine.org/). One dataset collected 19 patients, including 4 stage 4 and 1 stage 4S (Albino et al., 2008). The other one dataset collected 117 patients, including 27 stage 4 and 57 stage 4S (Asgharzadeh et al., 2006). All gene expression profiles were analyzed by microarray technology. We selected genes that were expressed with a significant difference (P < 0.01) between the two stages.

The validated genes were further analyzed the relationships between the gene expression and patients’ overall survival rate using survival package. Significant difference criterion was determined by a P-value <0.05.

Tumour-associated immune cell infiltration analysis

Tumour infiltrating leukocytes (TILs) are an integral component of the tumour microenvironment (TME) and correlate with prognosis and response to treatment. CIBERSORT (Chen et al., 2018) (https://cibersort.stanford.edu) is a versatile computational method used to quantify and distinguish 22 leukocyte subsets from bulk tissue gene expression profiles. This database provided the gene expression matrix of 22 immune cells as a background correction file and corresponding R script. We explored the differences in the infiltrating immune cells between the two stages by mapping and integrating the matrix to GEO gene expression matrices by introducing the R script downloaded from CIBERSORT. Statistical significance was determined by a P < 0.05.

Validation of hub genes

We obtained two significantly upregulated genes by combining the GEO database and the Oncomine database, neuromedin U (NMU) and neurotensin (NTS), between stage 4S and stage 4 in neuroblastoma. To validate the function of the two molecules, we performed in vitro experiment using the RNA interfering (RNAi) method.

Cell culture

The human neuroblastoma cell line SK-N-BE(2) was donated by Chenjie Tang, which was purchased from Zhong Qiao Xin Zhou Biotechnology Corporation (Shanghai, China). The cells were cultured in F12k & MEM (v/v = 1:1) medium with 10% fetal calf serum (Gibco, Thermo Fisher Scientific, USA), 100 U/ml penicillin, and 100 U/ml streptomycin in 5% atmospheric CO2 at 37 °C. Cells were cultured in 6-well plates with 2.5 × 105 cells/well.

siRNA transfection

Chemosynthetic siRNAs of NMU and NTS were purchased from Ribobio Techonology Corporation (Guangzhou, China). The target sequences for the NMU gene were: 5′-CAAUGUUGUAAAUGUUCAAUU-3′ (sense), 5′-UUGAACAUUUACAACAUUGUU-3′ (antisense), and for the NTS gene were: 5′-GGAAGAUGACUCUGCUAAAUG-3′ (sense), 5′-UUUAGCAGAGUCAUCUUCCAA-3′ (antisense). Negative control sequences were: 5′-GCUGUCGACUGAGUUAGAGGA-3′ (sense), 5′-GUCCAUCUCAUAACACAGCUG-3′ (antisense). siRNAs were diluted and transiently transfected with lipofectamine 2,000 in cells (Invitrogen, Thermo Fisher Scientific, USA) in 6-well plate (Corning, USA) for a final concentration of 100 nM per well once the cell monolayer reached 80%–90% in thebase of each well.Cells were harvested after 48 h of transfection for Transwell assay and wound-healing assay. Experiments were repeated in triplicate.

Western blot

To validate the interfering effect, we used a Western blot assay to detect the expression of NMU and NTS before and after transfection of siRNA. Rabbit polyclonal anti-NMU (20 kDa, Lot#: ab229782) antibody and rabbit polyclonal anti-NTS (20 kDa, Lot#: ab233107) antibody were purchased from Abcam (USA); GAPDH rabbit monoclonal antibody(36 kDa, Lot#: AF1186) and HRP-labeled goat anti-rabbit IgG (H+L, Lot#: A0208) were purchased from Beyotime (Shanghai, China). The optimal dilutions of the primary antibodies and secondary antibody were determined by anti-NMU and anti-NTS as 1:1,000, anti-GAPDH as 1:3,000, and the secondary antibody as 1:500, respectively.

Cells were trypsinized and washed twice with phosphate buffer saline (PBS, Hyclone, USA). Protein lysate was obtained by RIPA lysis buffer (Beyotime, Shanghai) according to the manufacturer’s instructions. Protein lysates (50 µg in each well) of identical quality were loaded and separated by polyacrylamide gel electrophoresis and transferred to the polyvinylidene fluoride membrane (Bio-Rad, Hercules, CA, USA). Primary antibodies were incubated in 4 °C for 16 h, and secondary antibodies were incubated in room temperature for 1 h. Finally, ECL chemiluminescence reagent (Millipore, USA) was used to detect the expression of protein.

Transwell assay

Cells were harvested and added to a 24-well Transwell plate (Corning, USA) with 1 ×104 cells per well after 48 h of transfection. Cells were cultured for 24 h and then stained with 1% crystal violet solution to detect the invading cell numbers under the microscope. Experiments were repeated twice, and ten fields of photos taken under microscope were taken for each group.

Wound-healing assay

The cell monolayer in the 6-well plate was scraped by 10 µl pipette tip, then washed twice using PBS (Hyclone, USA). Three fields of photos were taken for each group at 200× resolution by microscope during the initial (0 h) gap width.Three addition fields of photos were taken for each group after 48 h of cell cultivation to assessthe residual gap width. The width counts were calculated by imagePro Plus software (Media Cybernetics Inc., USA). The change of gap width was determined by | width0h-width48h |; three fields of photos under microscope were adopted for each group.

Co-expression relationships of hub genes and immune-related genes

Pearson test was used to analyze the relationships of hub genes expression and infiltrated immune cells, thereby to detect the potential function of these critical genes in neuroblastoma development. Correlation coefficient was determined by a r > 0.5 and P <0.05.

Results

Gene ontology analysis revealed that genes were enriched in aggressive biological progression

We obtained 309 and 1,103 DEGs from the GSE120572 and GSE73517 datasets, respectively, when comparing INSS stage 4S and stage 4. Of these, 216 intersected DEGs were further screened from the two datasets (Fig. 1A).

Figure 1 Gene Ontology enrichment analysis for biological processes (GO_BP) of significant differentially expressed genes (DEGs), which were compared between stage 4 and 4S.

(A) 216 DEGs were intersected from the GSE120572 and GSE73517 datasets. (B) GO enrichment analysis showed aggressive characteristics of these DEGs that implicating in tumor biological processes.

Consequently, GO analysis revealed that BP terms of the 216 DEGs were enriched in positive regulation of the development process, cell adhesion, signal transduction, and negative regulation of cell death, suggesting that these genes are involved in aggressive tumour development (Fig. 1B, Table S3).

NMU and NTS were identified as elevated hub genes in stage 4

We constructed protein-protein interactions (PPI) networks in the STRING database to learn more about the function of the identified genes. Their networks were subsequently visualized by Cytoscape (Fig. 2A, Fig. S1). The top 25 hub genes were screened out with the Degree method using CytoHubba (Fig. 2B, Table S4).

Figure 2 Screening and validation of hub genes between stage 4S and stage 4.

(A) Workflow of hub genes screening. (B) Visualized network of Hub genes by Cytoscape, 25 hub genes were included by Degree method. (C–D) NMU and NTS were further determined to be significanly elevated in the Oncomine database. (E–F) NMU and NTS were inversely correlated with neuroblastoma patients’ overall survival rate.

We then extracted expression values of the 25 genes in stage 4S and stage 4 from the Oncomine database, respectively, and compared the differences between the two stages. Results showed NMU and NTS were significantly elevated in stage 4, without obvious differences being found between the two stages for the remaining 23 genes (Figs. 2C, 2D).

Combining data from TARGET database, we found that patients’ overall survival rate in high NMU expression group was significantly lower than that in low expression group (P = 2.25e−06). NTS presented similar action (P = 0.019) (Figs. 2E, 2F).

Significant tumour-associated lymphocytes infiltration occurred in stage 4

By integrating the 22 immune cells basic gene expression matrix, which as a background gene expression file, to GSE120572 and GSE73517, we found there was no significant (P > 0.05) immune cell infiltration in INSS stage 4S for any of the two datasets. Conversely, obvious infiltration of tumour-associated macrophages type 2 (TAM-M2) and regulatory T cells (Tregs) was determined in stage 4, for both datasets (Figs. 3A–3D).

Figure 3 Immune cells infiltrating analysis.

(A–B) The proportions of 22 infiltrating immune cells in stage 4 of neuroblastoma of the GSE120572 dataset displayed by box plot and heatmap. (C–D) The proportions of 22 infiltrating immune cells in stage 4 of neuroblastoma of the GSE73517 dataset displayed by box plot and heatmap.

NMU and NTS promote cell proliferation and invasion of SK-N-BE(2)

The results of the Western blot assay showed obvious interference of the target genes (Figs. 4A, 4B). The results of the transwell assay showed that the invaded cell number in si-NMU and si-NTS group were significantly less than the corresponding negative control (P < 0.0001, t = 12.81; P < 0.0001, t = 8.714), respectively (Figs. 4C–4E, 4F–4H). The results of the wound-healing assay revealed the interference of NMU and NTS led to a significant migration distance less than the negative control (P = 0.0003, t = 11.69; P = 0.0002, t = 13.04), respectively (Figs. 4I–4J, 4K–4L, 4M). Our results suggested that NMU and NTS could facilitate the invasion and migration of SK-N-BE(2).

Figure 4 The functional examination of NMU and NTS in SK-N-BE(2) cell.

(A–B) Western blot suggested that siRNA of NMU and NTS could significantly decreased protein expression compared to negtive control group. (C–E) Transwell assay showed downregulated NMU reduced invaded cell number compared to negative control. (F–H) Transwell assay showed downregulated NTS reduced invaded cell number compared to negative control. (I–L) Compared to negative group, si-NMU group or si-NTS showed a slower migration speed from 0 h to 48 h. (M) Wound healing assay revealed that downregulated NMU or NTS reduced tumour cell migration speed compared to negative control.

NMU and NTS were significantly correlated with immune-related genes

Results of genes co-expression analysis showed that expression of NMU was positively associated with CD4, CD25, FOXP3, NFAT, IL-4 and IL-13 (Figs. 5A–5F, 5M–5R). NTS presented positive relationships with CD163, CD206, IL-10, TGF-β, STAT3 and STAT4, both in GSE120572 and GSE73517 datasets (Figs. 5G–5L, 5S–5X).

Figure 5 The genes co-expression correlation analysis for NMU and NTS in the GSE120572 and GSE73517 datasets.

(A–F, M–R) NMU was positively associated with CD4, CD25, FOXP3, NFAT, IL-4 and IL-13. (G–L, S–X) NTS presented positive relationships with CD163, CD206, IL-10, TGF-β, STAT3 and STAT4, both in the GSE120572 and GSE73517 datasets.

Discussion

The tumour microenvironment is the cellular environment comprising tumour cells, immune cells, stromal cells, and the extracellular matrix. The TME plays three important roles in stimulating the development of tumours by providing a growing environment for tumours, disabling the efficacy of anti-tumour drugs, and assisting tumour cell evasion from immune surveillance (Kerkar & Restifo, 2012; Patel et al., 2018; Potter, 2007). The mechanisms underlying the interaction between immune cells and neuroblastoma cells have been studied, but many questions remain unanswered, including the identification of key genes and how they to modulate TME. INSS stage 4S neuroblastoma is a special category of metastatic disease that may regress spontaneously. It has been shown to have a favourable outcome when compared to INSS stage 4 (Nickerson et al., 2000; Schleiermacher et al., 2003); thus, comparing the molecular differences between the two stages may provide some explanation for tumour progression. We demonstrated the significant infiltration of Tregs and TAM-M2 in stage 4 cases from two datasets, without obvious immune cell infiltration in stage 4S.

Tregs are immune cells characterized by immunosuppression and hypoergia (Campbell, 2015) and can restrain the function of effector cells and attenuate the body’s immune response, accelerating tumour angiogenesis and development (Sakaguchi et al., 2020). With single-cell sequencing analysis, Zheng et al. (2017) proved that non-functional CD8+ T cells and Tregs were enriched in liver cancer tissues, suggesting that Tregs may inhibit CD8+ T cell activity to promote liver cancer progression. Further, tumour-associated macrophages (TAMs) participate in the process to facilitate tumour immune escape through polarization (Hao et al., 2012). Studies have shown that tumour-derived IL-4, IL-5, IL-6, IL-10, and IL-13 can stimulate TAM polarization towards TAM-M2 (Martinez et al., 2008). Further, cytokines including IL-4, TGF-β, IL-17, VEGF, PDGF, M-CSF, and MMPs secreted by TAM-M2 (Coffelt, Hughes & Lewis, 2009) are involved in tumour invasion, metastasis, angiogenesis, and lymphangiogenesis. IL-10 derived from TAM-M2 has been implicated in the suppression of NK cells and cytotoxic T lymphocytes (Benner et al., 2019), thereby reducing the tumour cells to be killed. The accumulation of Tregs and TAM-M2 is a significant event in tumour development.

It is difficult to determine the underlying causes of the changes in TME between INSS stage 4S and stage 4. We sought to determine the molecular alterations influencing TME development and found that neuromedin U (NMU) and neurotensin (NTS) are hub genes elevated in stage 4 compared to stage 4S. In addition, we compared the gene expression differences between MYCN amplified and MYCN non-amplified groups in these stages, however, NMU and NTS were not found in the differentially expressed genes. This suggested that NMU and NTS may not be driven by MYCN amplification (The results were not presented). Studies that have linked NMU to cancer indicate that NMU is significantly upregulated in most malignancies, including neuroblastoma (Martinez et al., 2017; Przygodzka et al., 2019). NMU upregulation in the tissues and cell lines of malignancies correlated with increased invasiveness and metastatic potential of tumour cells and advanced stages, such as head and neck carcinoma, pancreatic tumour and lung cancer (Tokumaru et al., 2004; Wang et al., 2016; Yang et al., 2018). Lebedev et al. (2019) demonstrated that the increased expression of NMU was associated with poor prognosis in neuroblastoma. Similarly, the oncogenic action of NTS has been described in different cancers with effects at each step of progression (Ouyang et al., 2017). For example, NTS promotes glioma cell invasiveness through NTSR1 in vitro and in vivo (Ouyang et al., 2015). NMU and NTS are known to exert functions by binding to their relevant receptors and activating the corresponding signalling pathways to directly or indirectly accelerate tumours into malignant characteristics.

The influence of NMU and NTS on the tumour microenvironment has not been reported. NMU and NTS receptors, which belong to G-protein-coupled receptor (GPCR) families, were found to be expressed on the membrane of immune cells including T cells, B cells, macrophages, dendritic cells, mast cells, and on tumour cells (Hedrick et al., 2000; Kim et al., 2006; Magazin et al., 2004; Saada et al., 2012). Johnson et al. (2004) confirmed that the binding of NMU to NMUR increased the synthesis and secretion of IL-4, 5, 6, 10, and 13 in CD4+ T cells, and may be mediated as follows: activated phospholipase C (PLC) induced by G protein hydrolysed phosphoinositide 4,5-bisphosphate (PIP2) into diacylglycerol (DAG) and inositol 1,4,5 Tris-phosphate (IP3); IP3 subsequently elicited Ca2+ release from the endoplasmic reticulum, which resulted in calmodulin binding, stimulation of calcineurin, and dephosphorylation of NFAT family members, thereby facilitating the production of ILs (Johnson et al., 2004; Park et al., 2012; Smith-Garvin, Koretzky & Jordan, 2009). The binding of IL-4 and IL-13 to their receptors is linked to macrophage polarization to the M2 type via activation of the JAK-STAT pathways (Martinez et al., 2008; Mosser & Edwards, 2008). Kim et al. (2006) suggested that NTS immediately mediated the stimulation of the JAK-STAT1 pathway in macrophages. The JAK-STAT pathway is reported to be pivotal for TAM polarization to the M2 type (Wilson, 2014). IL-10 and TGF-β are powerful stimulants secreted by TAM-M2 and can transform CD4+ T cells into Tregs (Liu et al., 2013; Shen et al., 2015; Zhou et al., 2017).

In this study, we found NMU was significantly correlated with the expression of CD4, CD25, FOXP3, NFAT, IL-4 and IL-13, suggesting NMU may perform positive action in Tregs activation and engage in TAM-M2 polarization related cytokines release from CD4+ T cells. Meanwhile, NTS presented positive relationships with CD163 and CD206, which are the molecular biomarkers of TAM-M2, reminding us that NTS may involve in polarization of tumour infiltrating macrophages to M2 subtype. Co-expression of IL-10, TGF-β, STAT3 and STAT4 partially demonstrated NTS may facilitate M2 cells to release cytokines, thereby in return to strengthen CD4+ T cells activation. Moreover, results showed NMU and NTS were inversely correlated with neuroblastoma patients’ overall survival rate. These further provided us an evidence that NMU and NTS may influence neuroblastoma development by interfering its microenvironment.

Our data, and the results of previous studies, suggest that NMU and NTS may mediate the interaction of cancer cells and immune cells and may influence the progressionof neuroblastoma (Fig. 6), however further in vitro and in vivo experiments should be undertaken to confirm this speculation.

Figure 6 The possible mechanism of interactions between tumor cells and tumor-associated leukocytes.

Neuroblastoma cells may synthesize and secret NMU to influence the generation of IL-4 and IL-13 by CD4+ T cells. Meanwhile, NTS, which is also secreted by tumor cells, combine with IL-4 and IL-13 to stimulate the polarization of TAM to M2. In addition, IL-10 and TGF-β secreted by excessively activated M2 could stimulate CD4+ T cells to transform into Tregs. These complicated interaction networks thus in return to accelerate neuroblastoma development.

Conclusions

Our results indicate that NMU and NTS may play roles in the deterioration of the neuroblastoma microenvironment and may provide evidence for future treatments.

Supplemental Information

Supplemental Information 1 The clinical characteristics of the GSE120572 and GSE73517 datasets

Click here for additional data file.

Supplemental Information 2 Clinical characteristics of neuroblastoma patients in TARGET database, including patients’ survival status

Click here for additional data file.

Supplemental Information 3 Gene ontology analysis

Results of biological process enrichment analysis for differentially expressed genes, which were intersected from datasets GSE120572 and GSE73517.

Click here for additional data file.

Supplemental Information 4 Hub genes screening

25 hub genes screened by CytoHubba ranked by Degree method.

Click here for additional data file.

Supplemental Information 5 The PPI network construction

The protein to protein interaction analysis of 216 intersected genes.

Click here for additional data file.

Supplemental Information 6 Result of protein expression by interfering NMU and NTS

The protein as internal reference for western blot assay was GAPDH, which the predicted molecular weight is 36 kDa, and the predicted molecular weight of NMU and NTS are both 20 kDa.

Click here for additional data file.

Supplemental Information 7 Raw data of invaded cell numbers for Figs. 4B and D

Raw data of invaded cell numbers by interfering expression of NMU and NTS, respectively.

Click here for additional data file.

Supplemental Information 8 The raw data of Wound-Healing assay

The relative interval width changes between initial width(0h) and residual(48h).

Click here for additional data file.

Additional Information and Declarations

Competing Interests

Author Contributions

DNA Deposition

Data Availability

The authors declare there are no competing interests.

Daheng Yang and Xianwei Zhang performed the experiments, analyzed the data, prepared figures and/or tables, and approved the final draft.

Zheqian Li analyzed the data, prepared figures and/or tables, authored or reviewed drafts of the paper, and approved the final draft.

Fei Xu and Hongbing Chen conceived and designed the experiments, authored or reviewed drafts of the paper, and approved the final draft.

Chenjie Tang analyzed the data, prepared figures and/or tables, donated neuroblastoma cell line SK-N-BE(2), and approved the final draft.

The following information was supplied regarding the deposition of DNA sequences:

The neuromedin U sequence described here is available at GenBank: NC_000004.12.

The neurotensin sequence described here is available at GenBank: NC_000012.12.

The following information was supplied regarding data availability:

The datasets generated and analyzed are available at Figshare: Yang, Daheng (2021): Neuromedin U and neurotensin may promote the development of the tumour microenvironment in neuroblastoma. figshare. Dataset. https://doi.org/10.6084/m9.figshare.14554089.v1.

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
