# Peer review of "Neuromedin U and neurotensin may promote the development of the tumour microenvironment in neuroblastoma"

_PeerJ, doi:10.7717/peerj.11512_

## Round 0.1 · original submission · Major Revisions

Please revise your manuscript according to the reviewers' comments.

·

Basic reporting

The article must be written in English but the authors should improve it avoiding typing errors.
The authors included a sufficient introduction and background with explanation of own hypothesis. However not revelant literature on Neuroblastoma was referenced. I suggest to cite Luksch R et al. 2016.
The article should be improved in clarity and should be implemented in Methods and Results section with more detailed description of data.
The hypothesis was not fully sustained by overwhelming results. The authors should provide additional experiments to adequately support own hypothesis.

Experimental design

The authors define well the research question, however several past studies provided Neuroblastomas gene expression profiles resulting in a number of gene expression
signatures being used to evaluate the patient’s risk (De Preter et al., 2010; Oberthuer et al., 2010). Unfortunately, findings derived form these studies have never been translated into clinical practice.

The knowledge gap was identified and the study attempted to fill the gap but I suggest to tone down the wording on the functional consequences of authors' hypothesis.

The investigation seems to be conducted rigorously but not all experiments showed a high technical standard.

Methods were absolutely not described with sufficient information to be reproducible by another investigator.

Validity of the findings

The authors' hypothesis could have a good rationale, but the study needs to be improved in experimental design with adeguate Neruoblastoma risk stratification. It will add value to the literature providing novelty to the actual information on Neuroblastoma.

The data on which the conclusions are based are not robust and nedded to be improved.
The conclusions are appropriately stated and are connected to the original question investigated. However, the experiments provided do not support adequately the hypothesis.

Additional comments

In this study, Yang and co-workers analyzed the molecular differences in the tumour microenvironment between INSS stage 4S and stage 4 examining gene expression microarray data from two public datasets. Biological processes of differentially expressed genes (DEGs) were assessed by gene ontology enrichment analysis revealing that 216 DEGs were enriched in aggressive biological processes. The authors highlighted hub genes using STRING database and Cytoscape verifying gene expression in Oncomine database and found Neuromedin U and neurotensin elevated in stage 4 . To functionally validate in silico findings the authors performed transwell and wound healing assay to explore the function of the genes in neuroblastoma cell line SK-N-BE(2). Moreover, they analyzed leukocyte population infiltrating tumour showing a high infiltration of regulatory T cells and type 2 tumour-associated macrophages in stage 4 but not in stage 4S. The authors conclude from their data that neuroblastoma aggressiveness may be associated with the action of neuromedin U and neurotensin in modulating tumour microenvironment and suggest that this information might be used for the development of targeted therapies.

Specific comments:
1. A major limitation of this study is the relatively low number of cases in the two groups of two datasets. Taking also into account that the study has as starting point a re-analysis of public microarray datasets there is a certain risk of over-fitting and over-interpretation of the results. I strongly suggest to validate the results in an independent cohort. This task might be achieved optimally by analyzing another microarray dataset. An example is E-MTAB-161 dataset containing information of 504 NB smaples. Validation of the data in an independent cohort would substantially strengthen the significance of the results.

2. In the gene expression microarray dataset examined, the authors extrapoled the gene expression data of stage 4S and stage 4. However, since the high NB tumor heterogeneity as well as clinical outcome, to compare NB patients stage 4 versus 4S could have high risk again in over-interpretation of the results. Do the authors take into account only NB patients stage 4S with disease regression? It is note that patients stage 4S have different outcome according also MYCN amplification. It would be very interesting to repeat the analysis using stratification risk groups according to the High Risk Neuroblastoma Study 1.7 of SIOPEurope (SIOPEN) specifications considering HR group patients stages 2, 3, 4, and 4S with MYCN gene amplification or stage 4 samples with an age at diagnosis over 12 months of age; ‘low/intermediate- risk’ (LIR) group the remaining patients. The study would benefit from integrating the performed analysis by adding the Kaplan–Meier analysis on Neuromedin U and neurotensin and stage risk.

Minor revision
1. In general, the Methods section would be implemented with more detailed on procedure protocols. An example in “6.3 Western blot” control antibody used is lacking as well as information on antibodies incubation time, protein extraction and gel running. The authors would make substantial changes to the manuscript in writing and editing.
2. Same comment for Methods section. The Results section should be expanded and described more comprehensively.
3. In paragraph 2 of Results, the authors explored protein-protein interaction of 216 DEGs. However, the authors should provide the results of the analysis before showing the top 25 hub genes.
4. Paragraph 3 of Results. The study would benefit from sharpening the hypotheses on NB tumour progression. I suggest changing the last sentence since NB tumour progression don’t’ follow the roles of Vogelstein “multistep tumour model”. It is not very clear as CIBERSORT works, please provide more detailed since from this analysis the authors based own speculations.
5. Paragraph 4 of Results. The authors showed western blot of two proteins of interest. However, the two proteins are both of 20 kDa. Did the authors incubate the antibodies on the same membrane? If, yes, they should run protein lysate so that they can incubate the full membrane with two antibodies in parallel, without stripping. Moreover, the authors should show the effects of silencing on RNA. I suggest to perform RT-PCR.

6. The authors should support own hypothesis on the role of Neuromedin U and neurotensin in tumour aggressiveness with additional functional tests, such as clonogenic assay.

7. Conclusion. I suggest to tone down the wording on the functional consequences of genes of interest, and even more on their potential significance as therapeutic targets.

8. Written English could be improved and typing error had to be corrected. Some examples where the language have to be improved include “wound healing assay to detect the funtion of the genes in neuroblastoma cell line SK-N-BE(2) in abstract section, change funtion in function; lines 134 add “t” to scrip!!

Reviewer 2 ·

Basic reporting

In this manuscript, Yang and coworkers describe the molecular differences in the tumour microenvironment (TME) between Neuroblastoma INSS stage 4S and stage 4. The work is presented quite well and shows that Neuromedin U (NMU) and neurotensin (NTS) were elevated in stage 4. The authors also demonstrate that NMU and NTS promoted the proliferation and invasion of SK-N-BE(2) cell. Lastly, the authors showed a high infiltration of regulatory T cells and type 2 tumour-associated macrophages in stage 4 but not in stage 4S. However, there is no direct link here to show a relationship between NMU /NTS and TME. Overall this is an interesting piece of work, but much work needs to be done to improve the robustness of the findings.
1. Please could the authors check the manuscript throughout for grammatical errors as there are sentences which do not make sense.
2. The introduction needs to be improved. The authors should describe better lines 93-97 to provide more justification for the study.
3. The figures need to be improved:
- Figure 1. Could the authors label A and B for the upper and lower panel?
- Figure 2. The authors state that NMU and NTS were significantly elevated in stage 4 however no statistics are shown on the bar chart.
- Figure 4. There is no correlation between figure legend and labelled panel, please add the western blot images in the figure. The wound healing assay is incomplete. Have the authors measured the migration distance? They need to report the statistics of this analysis.

Experimental design

In general, the Materials and Methods section is not well described the authors should provide more detailed on procedure protocols. Some examples are reported below.
1. In 6. Validation of hub genes section, the author should describe better how they select NMU and NTS for further validation
2. In 6.2 siRNA transfection section more detail is required on how this was performed and timing of experiments.
3. In 6.3 Western blot section more detail is required in regard to how the proteins were obtained and how the western blot was performed.
4. In 6.5 Wound healing assay section please clarify and provide more detail in regard to how the microscopy was conducted and how the wound healing was measured.

Validity of the findings

As for the Materials and Methods section, the Results need to be better described. The authors state that NMU and NTS may be implicated in the modulation of TME and neuroblastoma development, however, this hypothesis was not fully supported by results.
1. In supplementary Table 2 the authors reported the 25 hub genes screened by CytoHubba ranked by Degree method, could they also report the genes obtained after combination with the Oncomine database? Furthermore, could the authors comment on why NMU and NTS were selected for further validation?
2. In “3. Significant tumour-associated lymphocytes infiltration occurred in stage 4” section it is shown that tumour-associated macrophages type 2 (TAM-M2) and regulatory T cells (Tregs) were significantly infiltrated in stage 4, but there is no direct link here with NMU and NTS expression. The authors should discuss the possible causes this.
3. The “4. NMU and NTS promote cell proliferation and invasion of SK-N-BE(2)” section needs to be improved. First, cell proliferation and migration are different concepts. Wound healing experiments can indicate the cell migration while the BrdU assay, MTT assay or Ki67 may be used to analyze cell proliferation. Furthermore, no statistical tests have been conducted to support the effect of downregulation of NMU and NTS on cell migration.

Reviewer 3 ·

Basic reporting

This paper used open data of gene expression to explore why stage 4 vs. stage 4S neuroblastoma are different. The authors chose two out of the 25 hub genes that were differentially expressed at higher levels in stage 4 NB and confirmed their association with invasiveness of the BE2 cells, a neuroblastoma cell line with MYCN amplficiation.

A major issue is that MYCN amplification is a very strong driving factor of pathogenesis and worse prognosis in neuroblastoma but this factor was not considered nor analyzed.

Minor points:

There are some typo's in the text

Abstract: "Neuromedin U (NMU) and neurotensin (NTS) were verified to be elevated in stage 4," -- Please specify how they were verified

Fig. 4. Panels B and D: The authors may mean inva"d"ing cell number in the Y axis.

Line 250: "Determining the reasons underlying these changes in TME from INSS stage 4S to stage 4 is difficult" -- Most clinical cases of stage 4 neuroblastoma are not changed from Stage 4S neuroblastoma. They are probably different diseases with different biology. This sentence should be revised.

Experimental design

Lines 101-105: Please specify why these two datasets were chosen. Please specify which kind of gene expression platform was used and which kind of data (e.g. microarray or whole transcriptome) was analyzed. Please cite publications related to these datasets. Please confirm how many tumor samples were obtained before chemotherapy. Please describe more about the clinical background of these datasets, e.g. age, stage, treatment protocol, country/region, etc.

Line 110: Please explain what is |logFC| and why this criterion was chosen.

Line 126: Please explain how the authors perform the Oncomine analysis. Are the results of Oncomine using the two datasets or different datasets?

Validity of the findings

Lines 195-197: Please explain more about why NMU and NTS were identified or chosen as the major genes to be reported. Were they the only DE genes that were
A Figure of the workflow to identify these genes can be very helpful.

Figure 2A, right panel: Why are NMU and NTS labeled in pink? What are the differences between these two genes and the other genes? What's the difference between the left and right panels?
Figures 2B-2C: What does the scale mean? What do the error bars mean? Are these two genes the top two genes with the highest relative expression level in stage 4 NB in the Oncomine database? How big was the difference?
Please also revise Figure Legend to be more clear.

Figures 3A-3B. Which stage are these tumors?
Figures 3C-3D. Why were all tumors shown are stage 4? Can the authors compare stage 4 vs. stage 4S?

Additional comments

Stage 4 neuroblastoma is a heterogeneous group. Because MYCN-amplified vs. MYCN-non-amplified neuroblastoma are the two prominent subgroups in stage 4 neuroblastoma, the authors are advised to further stratify or compare among MYCN amplified, MYNC non-amplified, and stage 4S tumors.

The workflow about how and why the two genes were chosen to be reported should be explained more clearly.

---

## Round 0.2 · Major Revisions

As the reviewer's suggestion, please provide a much more detailed analysis, preferably accompanied by dedicated experimental data, focused in supporting the proposed hypothesis.

Reviewer 2 ·

Basic reporting

The authors of the present study revised the manuscript according to the comments included in the peer review.

Experimental design

Regarding the comments for the Methods section, the authors have added in revised manuscript the missing information and have responded to all the suggestions and corrections included in the peer review.

Validity of the findings

Regarding the comments for the Results section I still have two points.

Minor point:
In Figure 4 (A-B) the western blot image shows an increase of proteins after silencing with siNMU and siNTS compared to the control siRNA and the housekeeping protein GAPDH contrary to what is stated in the figure legend and elsewhere in the manuscript. This should be revised.

Major point:
The authors have postulated that NMU and NTS influence the interaction of cancer cells and immune cells combining both literature data and immune cells infiltration analysis. However, the work is only descriptive and in several instances no mechanism was proposed to explain this observation. It would therefore be extremely helpful and advance the study considerably, if the authors could provide a much more detailed analysis, preferably accompanied by dedicated experimental data, focused in supporting the proposed hypothesis before publication can be considered.
In addition, the authors performed a functional study in which they evaluate the effects of NMU and NTS in the invasion and migration of neuroblastoma SK-N-BE(2) cells. The observation is interesting but not explained in the study. Authors should investigate and discuss this evidence in more detail.

Reviewer 3 ·

Basic reporting

The revised version is much more clear and informative about the methods and results. The issue of MYCN was mentioned in introduction (lines 265-266) but was not analyzed.

Figure 1 and 1A Legend: Please clarify that the 216 DEGs were compared between stage 4 and 4S.

Figure 2A and related text (e.g. Lines 335-336): Please further describe how did the Oncomine verification work -- Was it derived from microarray datasets? How many samples/patients? How many were stage 4/4S?

Experimental design

I suggest to add a summary table of Supplemental Table 1 to summarize the numbers of key clinical and genomic characteristics of the samples enrolled (e.g. between stage 4 vs. 4S, how many had MYCN amplification, were diagnosed at older age, and had the other genomic alterations [when available]?)

Validity of the findings

The effect of MYCN amplification was not assessed in this analysis. The DEG may be driven by MYCN amplification as this alteration was usually enriched in stage 4.
I suggest the authors to compare the the expression of the two genes between MYCN amplified vs. non-amplified samples among the stage 4/4S samples that were enrolled for analysis and present the result in a figure panel or a supplemental figure.

Lines 924-934: A concise review of the function of NMU and NTS in normal cells and/or cancers and their potential roles in NB should also be discussed.

Additional comments

Thank you for making the thorough revision.

---

## Round 0.3 · accepted · Accept

The authors addressed all the comments from the reviewers, so I suggest the manuscript can be accepted by PeerJ.

Reviewer 2 ·

Basic reporting

No comment

Experimental design

No comment

Validity of the findings

No comment

Reviewer 3 ·

Basic reporting

The authors have clarified that the 2 hub genes was not associated with MYCN amplification.

Experimental design

no comment

Validity of the findings

no comment

Additional comments

Thanks to the authors' efforts to make the revision. I would recommend that the paper is ready for publication.